# Motion Prior Distillation in Time Reversal Sampling for Generative Inbetweening

**Wooseok Jeon**[1]**, Seunghyun Shin**[2]**, Dongmin Shin**[1]**, Hae-Gon Jeon**[1]*

[1]Department of Artificial Intelligence, Yonsei University
[2]AI Graduate School, GIST

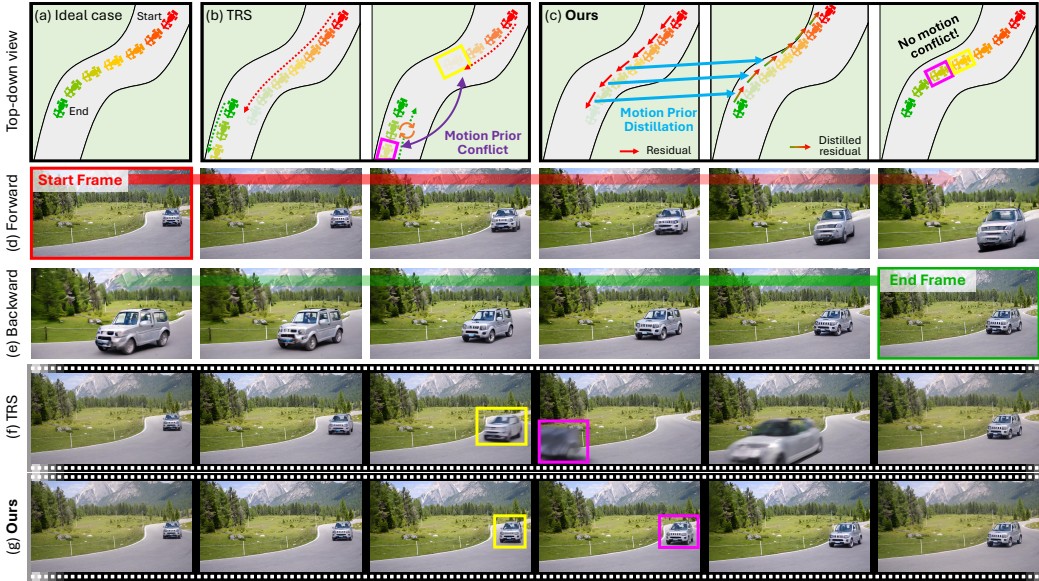

Figure 1: **Overview of the proposed motion prior distillation.** (a) Ideal case of generative inbetweening task. (b) Motion prior conflict in existing time reversal sampling method. (c) Our proposed motion prior distillation method. (d) A video generated by Stable Video Diffusion model conditioned on the start frame, and (e) conditioned on the end frame and temporally flipped. (f) A result from existing time reversal sampling method, showing ghosting artifact and reverse play due to motion prior conflict. (g) A result from our proposed method, showing temporally coherent motion.

## Abstract

Recent progress in image-to-video (I2V) diffusion models has significantly advanced the field of *generative inbetweening*, which aims to generate semantically plausible frames between two keyframes. In particular, inference-time sampling strategies, which leverage the generative priors of large-scale pre-trained I2V models without additional training, have become increasingly popular. However, existing inference-time sampling, either fusing forward and backward paths in parallel or alternating them sequentially, often suffers from temporal discontinuities and undesirable visual artifacts due to the misalignment between the two generated paths. This is because each path follows the motion prior induced by its own conditioning frame. In this work, we propose **Motion Prior Distillation (MPD)**, a simple yet effective inference-time distillation technique that suppresses bidirectional mismatch by distilling the motion residual of the forward path into the backward path. Our method can deliberately avoid denoising the end-conditioned path which causes the ambiguity of the path, and yield more temporally coherent inbetweening results with the forward motion prior. We not only perform quantitative evaluations on standard benchmarks, but also conduct extensive user studies to demonstrate the effectiveness of our approach in practical scenarios. Project page: https://vvsjeon.github.io/MPD/

---

*Corresponding author

# 1 INTRODUCTION

Recent advances in diffusion models have significantly improved the performance of image and video generation tasks. In particular, image-to-video (I2V) diffusion models (Blattmann et al., 2023a; Xing et al., 2024b; Bar-Tal et al., 2024; Yang et al., 2025b) demonstrate strong capabilities across diverse applications, as they can generate temporally coherent videos from a single conditioning frame. From a generative perspective, this progress has extended video frame interpolation to *generative inbetweening*, which aims to generate natural intermediate frames between two keyframes (See Fig. 1 (a)). However, I2V diffusion models are not directly applicable to bounded generation where both start and end frames serve as a dual-constraint.

To address this, recent studies have explored time reversal sampling, which employs temporally forward / backward denoising paths conditioned on the start / end frames during the iterative reverse denoising process (See Fig. 1 (b)). This can be categorized into two approaches, namely parallel and sequential, according to how these two paths are integrated. In the parallel approach (Feng et al., 2024; Wang et al., 2025b; Zhu et al., 2025), samples from the forward and backward paths are denoised simultaneously at each denoising step and then linearly interpolated to form the input for the next denoising step. In contrast, the sequential approach (Yang et al., 2025a) samples two denoising paths sequentially by inserting a single re-noising step between them.

However, simply connecting the two temporal paths does not guarantee a single coherent motion during the sampling process because each sample is obtained with the motion prior of its conditioning frame (See Fig. 1 (d) and (e)). In particular, as shown in Fig. 1 (e), a backward path initialized at the end frame tends to generate forward-looking sequences, instead of faithfully reconstructing historical frames. This forward-generation bias commonly arises from I2V models, which are trained to predict consecutive forward frames. As shown in Fig. 1 (f), the generated frames noticeably follow different routes and even disagree on the car's destination, which we refer to as a ***motion conflict*** between two temporal paths. This highlights that the fundamental challenge lies not merely in how to connect the forward and backward paths, but in how to align their conflicting motion priors induced by the forward-generation bias.

To this end, we aim to overcome this fundamental misalignment between two temporal paths by proposing a novel inference-time distillation approach, called **Motion Prior Distillation (MPD)**. Our key intuition is that the residual of the denoised estimates contains motion information induced by a given start frame. Inspired by this, during early denoising steps, our method distills the motion residual induced by the start frame into the backward path (See Fig. 1 (c)). Since our approach deliberately avoids denoising the end-conditioned path, we can drive the backward path to follow the time reversed motion residual of the forward path, thereby achieving bidirectional path alignment (See Fig. 1 (g)). This single path design effectively removes conflicting motion priors while preserving endpoint consistency, allowing two temporal paths to converge into coherent motions.

Through extensive evaluations, we demonstrate that our method consistently outperforms relevant methods, including existing time reversal sampling strategies. In addition, since conventional metrics are not fully capable of evaluating temporal coherence and human preference, we further conduct user studies to validate its robustness under practical scenarios in the presence of complex motion patterns and large temporal displacements.

# 2 RELATED WORKS

**Video frame interpolation.** Video frame interpolation (VFI) aims to synthesize intermediate frames between two input frames while maintaining spatial and temporal coherence (Lyu et al., 2024; Kye et al., 2025). Supervised methods (Bao et al., 2019; Niklaus & Liu, 2018; Park et al., 2020; Lei et al., 2023; Kong et al., 2022; Li et al., 2023; Huang et al., 2022; Lu et al., 2022; Reda et al., 2022) that rely on estimating optical flows have been practically adopted due to their robust performance and interpretable motion trajectories. However, errors in estimated flows often lead to failures, particularly under the scenes with occlusion or non-linear motion (Long et al., 2024). Recently, diffusion-based VFI methods (Danier et al., 2024; Voleti et al., 2022) have attempted to take advantage of the generative capabilities of diffusion models to improve the perceptual quality of interpolated frames. While these methods improve perceptual fidelity, their performance still degrades under large temporal displacements between two frames.

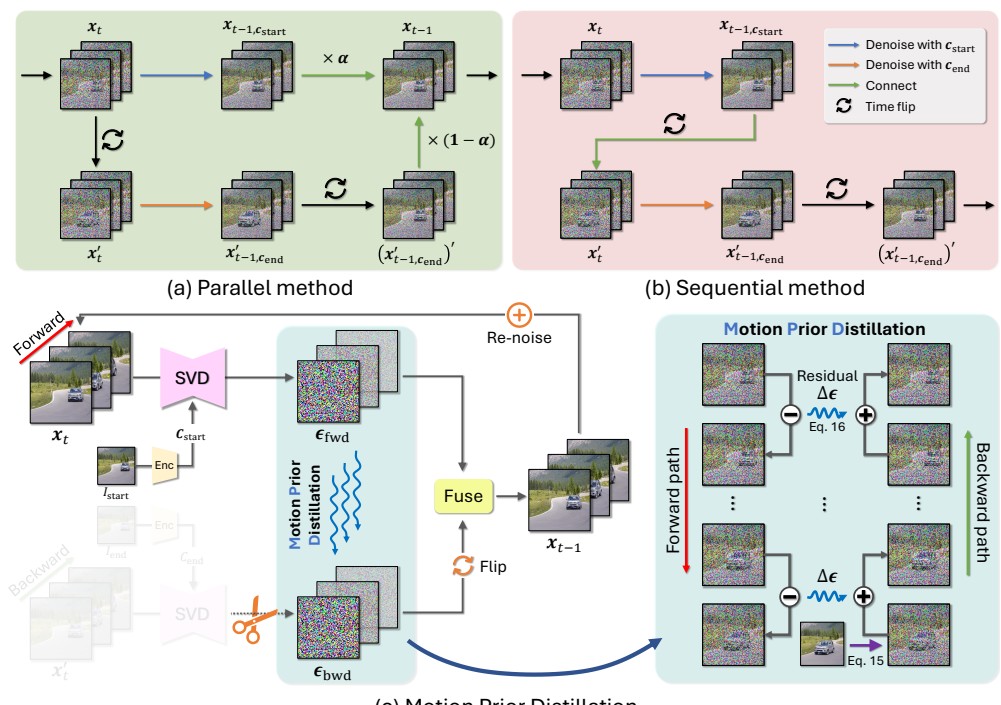

Figure 2: **Denoising process of the proposed motion prior distillation.** Existing time reversal sampling methods simply connect the two temporal paths either by (a) linearly fusing them or (b) alternatively denoising each path. (c) Our MPD is employed on time reversal sampling framework to distill forward motion prior into the backward path, thereby achieving motion alignment.

**Generative video inbetweening.** With the advancement of video diffusion models (Ho et al., 2022b;a; Blattmann et al., 2023b;a), VFI has broadened into *generative inbetweening*, which is interested in the set of semantically plausible interpolations. Some approaches (Jain et al., 2024; Xing et al., 2024a;b; Wang et al., 2025a; Zhang et al., 2025) train diffusion models to condition on two input frames for interpolation, yielding greater robustness to ambiguous and large motion where traditional methods have struggled. While effective, they typically require substantial training resources. Other approaches leverage pre-trained large-scale I2V diffusion models and achieve remarkable performances by incorporating new sampling techniques. TRF (Feng et al., 2024) proposes a time reversal sampling strategy that fuses forward and backward denoising paths in parallel, each conditioned on the start and end frames. Building on this strategy, GI (Wang et al., 2025b) enhances reverse motion fidelity by fine-tuning a diffusion model through rotation of temporal self-attention maps to generate temporally reversed frames. Similarly, FCVG (Zhu et al., 2025) proposes a method that injects line correspondences as frame-wise conditions to alleviate the ambiguity of inbetweening path. Meanwhile, ViBiDSampler (Yang et al., 2025a) introduces a new time reversal strategy that employs sequential sampling along forward and backward paths to achieve on-manifold generation of intermediate frames. However, all of them still operate with two independent motion priors from the start and end frames, so convergence to a single coherent trajectory is not guaranteed.

## 3 PRELIMINARIES

### 3.1 STABLE VIDEO DIFFUSION

We base our explanation on Stable Video Diffusion (SVD) (Blattmann et al., 2023a), which is widely adopted in time reversal sampling based methods. Specifically, SVD is a UNet-based latent video diffusion model built on EDM framework (Karras et al., 2022). At a reverse denoising step $t \in \{T, ..., 1\}$ with the noise level $\sigma_t$, the denoiser $\boldsymbol{D}_\theta$ predicts both the unconditional estimate $\hat{\boldsymbol{x}}_{0,\varnothing}$ and the conditional estimate $\hat{\boldsymbol{x}}_{0,\boldsymbol{c}}$ from the current noisy latent $\boldsymbol{x}_t$:

$$\hat{\boldsymbol{x}}_{0,\varnothing} = \boldsymbol{D}_\theta(\boldsymbol{x}_t; \sigma_t) \quad \text{and} \quad \hat{\boldsymbol{x}}_{0,\boldsymbol{c}} = \boldsymbol{D}_\theta(\boldsymbol{x}_t; \sigma_t, \boldsymbol{c}), \tag{1}$$

where $\boldsymbol{c}$ is the input condition. In EDM framework, the corresponding noise prediction model $\boldsymbol{\epsilon}_\theta$ and the score prediction model $\boldsymbol{s}_\theta$ have the following relationship with the denoiser $\boldsymbol{D}_\theta$:

$$s_\theta(\boldsymbol{x}_t; \sigma_t) = -\frac{\boldsymbol{\epsilon}_\theta(\boldsymbol{x}_t; \sigma_t)}{\sigma_t} = \frac{\boldsymbol{D}_\theta(\boldsymbol{x}_t; \sigma_t) - \boldsymbol{x}_t}{\sigma_t^2}. \tag{2}$$

To guide the sample toward the condition $\boldsymbol{c}$, the classifier-free guidance (CFG) (Ho & Salimans, 2021) mixes the unconditional estimate $\hat{\boldsymbol{x}}_{0,\varnothing}$ with the conditional estimate $\hat{\boldsymbol{x}}_{0,\boldsymbol{c}}$:

$$\hat{\boldsymbol{x}}_{0,\boldsymbol{c}} \leftarrow (1+w)\hat{\boldsymbol{x}}_{0,\boldsymbol{c}} - w\hat{\boldsymbol{x}}_{0,\varnothing}, \tag{3}$$

where $w \geq 0$ is a guidance strength. At each iteration, we can denoise the sample with Euler step, progressively denoising from Gaussian noise $\boldsymbol{x}_T$ to sample $\boldsymbol{x}_0$:

$$\boldsymbol{x}_{t-1} = \hat{\boldsymbol{x}}_{0,\boldsymbol{c}} + \frac{\sigma_{t-1}}{\sigma_t}\left(\boldsymbol{x}_t - \hat{\boldsymbol{x}}_{0,\boldsymbol{c}}\right). \tag{4}$$

In particular, I2V models take the initial starting frame condition as input and generate videos with its motion prior. To reflect both start and end frame conditions, time reversal sampling process involves denoising two temporal paths with each corresponding frame condition.

## 3.2 Time reversal sampling

**Parallel method.** As illustrated in Fig. 2 (a), parallel time reversal methods denoise the temporally forward/backward path conditioned on the start/end frame, and then fuse them to produce the intermediate frames (Feng et al., 2024; Wang et al., 2025b; Zhu et al., 2025). Let's denote $\boldsymbol{c}_{\text{start}}$ and $\boldsymbol{c}_{\text{end}}$ as the encoded latent conditions of the start and end frame, respectively. We can express the denoising step as follows:

$$\boldsymbol{x}_{t-1} = \alpha\boldsymbol{x}_{t-1,\boldsymbol{c}_{\text{start}}} + (1-\alpha)(\boldsymbol{x}'_{t-1,\boldsymbol{c}_{\text{end}}})' \tag{5}$$

$$\text{s.t.} \quad \boldsymbol{x}_{t-1,\boldsymbol{c}_{\text{start}}} = \hat{\boldsymbol{x}}_{0,\boldsymbol{c}_{\text{start}}} + \frac{\sigma_{t-1}}{\sigma_t}\left(\boldsymbol{x}_t - \hat{\boldsymbol{x}}_{0,\boldsymbol{c}_{\text{start}}}\right) \tag{6}$$

$$\text{and} \quad \boldsymbol{x}'_{t-1,\boldsymbol{c}_{\text{end}}} = \hat{\boldsymbol{x}}'_{0,\boldsymbol{c}_{\text{end}}} + \frac{\sigma_{t-1}}{\sigma_t}\left(\boldsymbol{x}'_t - \hat{\boldsymbol{x}}'_{0,\boldsymbol{c}_{\text{end}}}\right), \tag{7}$$

where $(\cdot)'$ indicates a temporal flip along the time dimension and $\alpha \in [0,1]$ refers to the interpolation weight. However, this method can suffer from off-manifold issues, where samples deviate from the learned data manifold. As a result, their linearly interpolated results often lead to oscillations and undesirable artifacts. Furthermore, they do not resolve the conflicting motion priors induced by the two conditions, so motion fidelity can still degrade.

**Sequential method.** An alternative approach adopts the sequential time reversal sampling strategy (Yang et al., 2025a). Instead of fusing two temporal paths in parallel, this method sequentially denoises the forward and backward paths as Fig. 2 (b). On-manifold generation can be achieved by inserting a single re-noising step before switching from the forward to the backward path:

$$\boldsymbol{x}_{t-1,\boldsymbol{c}_{\text{start}}} = \hat{\boldsymbol{x}}_{0,\boldsymbol{c}_{\text{start}}} + \frac{\sigma_{t-1}}{\sigma_t}\left(\boldsymbol{x}_t - \hat{\boldsymbol{x}}_{0,\boldsymbol{c}_{\text{start}}}\right), \tag{8}$$

$$\boldsymbol{x}_{t,\boldsymbol{c}_{\text{start}}} = \boldsymbol{x}_{t-1,\boldsymbol{c}_{\text{start}}} + \sqrt{\sigma_t^2 - \sigma_{t-1}^2}\,\varepsilon, \quad \varepsilon \sim \mathcal{N}(0,\boldsymbol{I}), \tag{9}$$

$$\boldsymbol{x}_{t-1} = \left(\hat{\boldsymbol{x}}_{0,\boldsymbol{c}_{\text{end}}} + \frac{\sigma_{t-1}}{\sigma_t}\left(\boldsymbol{x}_{t,\boldsymbol{c}_{\text{start}}} - \hat{\boldsymbol{x}}_{0,\boldsymbol{c}_{\text{end}}}\right)\right)'. \tag{10}$$

Unlike the parallel approach, this sequential structure maintains a more consistent and manifold-aligned path. Nevertheless, alternating two denoised paths results in conflicting motion priors, as each path relies on its own conditioning frame. This highlights the need to align two temporal paths without motion prior conflicts.

## 4 Method

Given a pair of two frames $\{I_{\text{start}}, I_{\text{end}}\}$, our goal is to align two temporal paths with both temporal coherence and visual fidelity. Fig. 2 (c) provides an overview of our method. To begin with, we recast the time reversal sampling process as an optimization problem to solve a bidirectional path misalignment problem. In this work, we present a simple yet effective approach called Motion Prior Distillation (MPD) which propagates a motion residual from a forward path into a backward path.

## 4.1 Motivation

The existing time reversal sampling methods can be interpreted as a sampling procedure in which each denoising path approximately minimizes the following loss function $\mathcal{L}$:

$$
\begin{aligned}
\mathcal{L}(\boldsymbol{x}; \theta, \boldsymbol{c}_{\text{start}}, \boldsymbol{c}_{\text{end}}, \sigma) &= \left\| \boldsymbol{\epsilon}_\theta(\boldsymbol{x}; \sigma, \boldsymbol{c}_{\text{start}}) - \boldsymbol{\epsilon}_\theta(\boldsymbol{x}'; \sigma, \boldsymbol{c}_{\text{end}})' \right\|_2^2 \\
&= \left\| \frac{\boldsymbol{x} - \hat{\boldsymbol{x}}_{0, \boldsymbol{c}_{\text{start}}}}{\sigma_t} - \frac{(\boldsymbol{x}')' - (\hat{\boldsymbol{x}}'_{0, \boldsymbol{c}_{\text{end}}})'}{\sigma_t} \right\|_2^2 \\
&= \frac{1}{\sigma_t^2} \left\| \hat{\boldsymbol{x}}_{0, \boldsymbol{c}_{\text{start}}} - (\hat{\boldsymbol{x}}'_{0, \boldsymbol{c}_{\text{end}}})' \right\|_2^2.
\end{aligned}
\tag{11}
$$

Here, the objective of Eq. (11) is to enforce the consistency between one path and a temporally reversed path in both directions by optimizing the noisy samples $\boldsymbol{x}$ as follows:

$$
\bar{\boldsymbol{x}} = \arg\min_{\boldsymbol{x}} \ \mathcal{L}(\boldsymbol{x}; \theta, \boldsymbol{c}_{\text{start}}, \boldsymbol{c}_{\text{end}}, \sigma),
\tag{12}
$$

where $\bar{\boldsymbol{x}}$ denotes the latent that minimizes the discrepancy between the two temporal paths. However, incompatible motion priors induced by two frame conditions introduce the ambiguity between the two denoising paths, especially in early denoising steps. Without resolving this problem, the loss $\mathcal{L}$ is optimized to make the misaligned path worse, causing implausible motions in generated videos, as illustrated in Fig. I. When there is a significant gap between two motion priors, we could observe unrealistic motions like reverse play. Note that various types of visual artifacts come from incompatible motion priors, which will be discussed in Sec. 5.2.

## 4.2 Bidirectional path alignment with Motion Prior Distillation

Since subsequent denoising steps primarily focus on restoring high-frequency details, previous works (Feng et al., 2024; Yang et al., 2025a) often fail to correct this misaligned trajectory. To resolve this issue, we introduce a single path sampling scheme that distills the motion prior induced by the start conditioning frame $\boldsymbol{c}_{\text{start}}$ into the backward path.

Here, our key intuition is that the forward motion residual $\Delta$ of the denoised estimates $\hat{\boldsymbol{x}}_{0, \boldsymbol{c}_{\text{start}}}$ contain useful motion information, which can be written as:

$$
\Delta \hat{\boldsymbol{x}}_{0, \boldsymbol{c}_{\text{start}}}{}^{(i)} := \hat{\boldsymbol{x}}_{0, \boldsymbol{c}_{\text{start}}}^{(i)} - \hat{\boldsymbol{x}}_{0, \boldsymbol{c}_{\text{start}}}^{(i-1)},
\tag{13}
$$

where $i \in \{2, ...N\}$ denotes the frame index, given $N$ frames. Then, using the relation between $\boldsymbol{D}_\theta$ and $\boldsymbol{\epsilon}_\theta$ in Eq. (2), the residual of noise from the forward path $\Delta \boldsymbol{\epsilon}_{\text{fwd}}$ is given as:

$$
\Delta \boldsymbol{\epsilon}_{\text{fwd}} = \frac{\Delta \boldsymbol{x}_t - \Delta \hat{\boldsymbol{x}}_{0, \boldsymbol{c}_{\text{start}}}}{\sigma_t},
\tag{14}
$$

where $\Delta \boldsymbol{x}_t = \boldsymbol{x}_t^{(i)} - \boldsymbol{x}_t^{(i-1)}$ represents the residual of the noisy sample $\boldsymbol{x}_t$. Now, given the encoded latent $\boldsymbol{z}_{\text{end}}$ of the end frame $I_{\text{end}}$, we initialize the first index of the backward denoised estimate $\hat{\boldsymbol{x}}'_{0, \boldsymbol{c}_{\text{end}}}{}^{(1)}$ with $\boldsymbol{z}_{\text{end}}$ as:

$$
\boldsymbol{\epsilon}_{\text{bwd}}^{(1)} = \frac{(\boldsymbol{x}'_t)^{(1)} - \boldsymbol{z}_{\text{end}}}{\sigma_t}.
\tag{15}
$$

Next, we reconstruct the backward noise residual $\boldsymbol{\epsilon}_{\text{bwd}}$ by cumulatively subtracting the forward noise residual from the initial backward noise $\boldsymbol{\epsilon}_{\text{bwd}}^{(1)}$:

$$
\boldsymbol{\epsilon}_{\text{bwd}}^{(i)} = \boldsymbol{\epsilon}_{\text{bwd}}^{(1)} - \sum_{k=2}^{i} \Delta \boldsymbol{\epsilon}_{\text{fwd}}^{(k)}.
\tag{16}
$$

It is noteworthy that we should ignore the end frame condition $\boldsymbol{c}_{\text{end}}$. Therefore, we reformulate Eq. (2) as follows:

$$
\hat{\boldsymbol{x}}'_{0, \boldsymbol{c}_{\text{start}}^*} = \boldsymbol{x}_t - \sigma_t \, \boldsymbol{\epsilon}_{\text{bwd}}.
\tag{17}
$$

Here, the reconstructed $\boldsymbol{\epsilon}_{\text{bwd}}$ from the residual of the forward noise $\Delta \boldsymbol{\epsilon}_{\text{fwd}}$ provides us with a denoised estimate $\hat{\boldsymbol{x}}'_{0, \boldsymbol{c}_{\text{start}}^*}$ from $\boldsymbol{c}_{\text{start}}^*$, which implies the flipped motion prior of $\boldsymbol{c}_{\text{start}}$. To curb off-manifold behaviors, we adopt CFG++ (Chung et al., 2025) following ViBiDSampler (Yang et al., 2025a).

---

**Algorithm 1** MOTION PRIOR DISTILLATION

---

**Input:** $\boldsymbol{x}_T \sim \mathcal{N}(0, \sigma_T^2 \boldsymbol{I})$, $\boldsymbol{z}_{\text{start}}, \boldsymbol{z}_{\text{end}}, \{\sigma_t\}_{t=1}^{T}$, hyperparameters $\lambda, k, \gamma$
**Output:** improved inbetweening result $\boldsymbol{x}_0$
1: $\boldsymbol{c}_{\text{start}}, \boldsymbol{c}_{\text{end}} \leftarrow \text{encode}(\boldsymbol{z}_{\text{start}}, \boldsymbol{z}_{\text{end}})$
2: **for** $t = T : (1-\gamma)T$ **do**
3:     **for** $j = 1 : k$ **do**
4:         $\hat{\boldsymbol{x}}_{0,\varnothing}, \hat{\boldsymbol{x}}_{0,\boldsymbol{c}_{\text{start}}} \leftarrow D_\theta(\boldsymbol{x}_t; \sigma, \boldsymbol{c}_{\text{start}})$                 ▷ denoise forward path with $\boldsymbol{c}_{\text{start}}$ (Eq. (1))
5:         $\Delta \boldsymbol{x}_t^{(i)} \leftarrow \boldsymbol{x}_t^{(i)} - \boldsymbol{x}_t^{(i-1)}$                                ▷ forward path residuals
6:         $\Delta \boldsymbol{x}_{0,\boldsymbol{c}_{\text{start}}}^{(i)} \leftarrow \hat{\boldsymbol{x}}_{0,\boldsymbol{c}_{\text{start}}}^{(i)} - \hat{\boldsymbol{x}}_{0,\boldsymbol{c}_{\text{start}}}^{(i-1)}$         ▷ forward denoised estimate residuals (Eq. (13))
7:         $\Delta \epsilon_{\text{fwd}} \leftarrow (\Delta \boldsymbol{x}_t - \Delta \hat{\boldsymbol{x}}_{0,\boldsymbol{c}_{\text{start}}})/\sigma_t$             ▷ forward noise residuals (Eq. (14))
8:         $\boldsymbol{x}_t' \leftarrow \text{flip}(\boldsymbol{x}_t)$                                      ▷ temporal flip
9:         $\epsilon_{\text{bwd}}^{(1)} \leftarrow ((\boldsymbol{x}_t')^{(1)} - \boldsymbol{z}_{\text{end}})/\sigma_t$           ▷ initialize first index of $\epsilon_{\text{bwd}}$ (Eq. (15))
10:       $\epsilon_{\text{bwd}}^{(i)} \leftarrow \epsilon_{\text{bwd}}^{(1)} - \sum_{k=2}^{i} \Delta \epsilon_{\text{fwd}}^{(k)}$               ▷ reconstruct $\epsilon_{\text{bwd}}$ (Eq. (16))
11:       $\hat{\boldsymbol{x}}_{0,\boldsymbol{c}_{\text{start}}^*}' \leftarrow \boldsymbol{x}_t - \sigma_t \epsilon_{\text{bwd}}$                ▷ reconstruct $\hat{\boldsymbol{x}}_{0,\boldsymbol{c}_{\text{start}}^*}'$ (Eq. (17))
12:       $\tilde{\boldsymbol{x}}_{0,\boldsymbol{c}_{\text{start}}} \leftarrow (1-\lambda)\hat{\boldsymbol{x}}_{0,\boldsymbol{c}_{\text{start}}} + \lambda(\hat{\boldsymbol{x}}_{0,\boldsymbol{c}_{\text{start}}^*}')'$       ▷ fuse two estimates (Eq. (18))
13:       $\boldsymbol{x}_{t-1} \leftarrow \tilde{\boldsymbol{x}}_{0,\boldsymbol{c}_{\text{start}}} + \frac{\sigma_{t-1}}{\sigma_t}(\boldsymbol{x}_t - \hat{\boldsymbol{x}}_{0,\varnothing})$       ▷ update with Euler step (Eq. (19))
14:       $\boldsymbol{x}_t = \boldsymbol{x}_{t-1} + \sqrt{\sigma_t^2 - \sigma_{t-1}^2} \varepsilon$                                 ▷ re-noise
15:     **end for**
16: **end for**

---

Consequently, the Euler step of SVD in Eq. (4) denoises the sample $\boldsymbol{x}_t$:

$$\tilde{\boldsymbol{x}}_{0,c_{\text{start}}} = (1-\lambda)\hat{\boldsymbol{x}}_{0,\boldsymbol{c}_{\text{start}}} + \lambda(\hat{\boldsymbol{x}}_{0,\boldsymbol{c}_{\text{start}}^*}')', \tag{18}$$

$$\boldsymbol{x}_{t-1} = \tilde{\boldsymbol{x}}_{0,c_{\text{start}}} + \frac{\sigma_{t-1}}{\sigma_t}(\boldsymbol{x}_t - \hat{\boldsymbol{x}}_{0,\varnothing}), \tag{19}$$

where $\lambda \in [0, 1]$ serves as the interpolation scale. Note that during this process, we intentionally do not denoise the temporally backward path with the end frame condition $\boldsymbol{c}_{\text{end}}$. This enables the direct transfer of the forward motion prior toward the end-frame constraint without introducing additional sources of misalignment. In addition, the proposed update in Eq. (19) can be seen as satisfying the proposed objective in Eq. (11) in a relaxed form. In our single-path update, the end-conditioned estimate is effectively replaced with the reconstructed estimate $\hat{\boldsymbol{x}}_{0,\boldsymbol{c}_{\text{start}}^*}'$ distilled from the forward motion prior. Thus, the loss that we define in Eq. (11) is simplified as:

$$\mathcal{L}(\boldsymbol{x}; \theta, \boldsymbol{c}_{\text{start}}, \boldsymbol{c}_{\text{start}}^*, \sigma) = \frac{1}{\sigma_t^2} \left\| \hat{\boldsymbol{x}}_{0,\boldsymbol{c}_{\text{start}}} - (\hat{\boldsymbol{x}}_{0,\boldsymbol{c}_{\text{start}}^*}')' \right\|_2^2. \tag{20}$$

By replacing the end frame condition $\boldsymbol{c}_{\text{end}}$ with $\boldsymbol{c}_{\text{start}}^*$, we can reduce the gap between the two temporal paths only with the start frame condition $\boldsymbol{c}_{\text{start}}$:

$$\bar{\boldsymbol{x}} = \arg\min_{\boldsymbol{x}} \mathcal{L}(\boldsymbol{x}; \theta, \boldsymbol{c}_{\text{start}}, \boldsymbol{c}_{\text{start}}^*, \sigma). \tag{21}$$

This reformulated loss shows that the backward path no longer introduces an independent motion prior; instead, it is aligned through the forward motion prior. This gives the denoiser $\boldsymbol{D}_\theta$ the opportunity to reconcile the original denoised path with its reconstructed counterpart within a timestep, producing a more stable trajectory.

## 4.3 PRACTICAL CONSIDERATIONS

Specifically, we disable MPD in later denoising steps. This choice follows the observation that diffusion sampling proceeds in a coarse-to-fine manner: early denoising steps with large $\sigma$ primarily represent global and low-frequency structure, whereas later denoising steps refine high frequency details (Rissanen et al., 2023; Kim et al., 2023; Wu et al., 2025). Complementary studies (Lee et al., 2025b;a; Park et al., 2025) further demonstrate that focusing guidance on these early steps yields better visual quality. Thus, modifying the motion prior is most effective when the trajectory is still being shaped globally. Motivated by these findings, we apply our method during the early denoising stage with additional re-noising steps $k > 0$ to steer the trajectory onto the correct direction. Then, we switch to existing time reversal samplers to enhance endpoint consistencies, which will be discussed in Secs. 5.3 and 5.4. More details of MPD are provided in Algorithm 1.

Table 1: **Quantitative comparison results on DAVIS and Pexels dataset.** We compare against six baselines. **Ours** + TRF and **Ours** + ViBiD refer to our method applied to the parallel and sequential time reversal sampling schemes, respectively. Best results are **bold**, and second-best are underlined.

| Method | DAVIS | | | | | Pexels | | | | |
|---|---|---|---|---|---|---|---|---|---|---|
| | LPIPS ↓ | FID ↓ | FVD ↓ | VB ↑ | VB++ ↑ | LPIPS ↓ | FID ↓ | FVD ↓ | VB ↑ | VB++ ↑ |
| FILM | 0.2946 | 55.160 | 1058.0 | 0.7978 | **0.9740** | 0.1157 | 43.935 | 761.60 | 0.8231 | 0.9734 |
| DynamiCrafter | 0.3158 | 46.739 | 678.92 | 0.7475 | 0.8735 | 0.2397 | 62.598 | 809.53 | 0.8211 | 0.9213 |
| TRF | 0.3127 | 56.894 | 674.31 | 0.7618 | 0.9352 | 0.2044 | 59.185 | 796.48 | 0.8008 | 0.9487 |
| GI | 0.2432 | 48.427 | 654.91 | 0.7747 | 0.9320 | 0.1114 | 47.990 | 476.93 | 0.8211 | 0.9566 |
| FCVG | 0.2347 | 38.997 | 621.82 | 0.7904 | 0.9353 | 0.1160 | 35.269 | 525.08 | 0.8245 | 0.9701 |
| ViBiD | 0.2492 | 39.883 | 559.49 | 0.7733 | 0.9387 | 0.1447 | 39.002 | 641.30 | 0.8130 | 0.9488 |
| **Ours** + TRF | **0.2212** | **34.910** | 612.17 | **0.7992** | 0.9330 | 0.1149 | **34.470** | 460.99 | **0.8503** | **0.9862** |
| **Ours** + ViBiD | 0.2220 | 37.241 | **527.05** | 0.7845 | 0.9474 | **0.1028** | 34.775 | **412.66** | 0.8235 | 0.9605 |

## 5 EXPERIMENTAL RESULTS

### 5.1 EXPERIMENTAL SETTINGS

**Evaluation dataset.** Following the previous works (Yang et al., 2025a; Wang et al., 2025b), we compare our method with relevant SOTA methods on two representative datasets. Specifically, we utilize 100 video-keyframe pairs from DAVIS dataset (Pont-Tuset et al., 2017), and 45 from Pexels [1]. To simulate typical inbetweening conditions where long-range temporal reasoning is required between sparsely spaced keyframes, those videos exhibit diverse and large motions such as driving, dancing, and so on.

**Implementation details.** We plug our method into both TRF (parallel) and ViBiD (sequential) building on SVD-XT model of SVD (Blattmann et al., 2023a) on a single NVIDIA RTX 4090 GPU. For the sampling process, we use the Euler scheduler with 25 timesteps with the default settings of SVD. Additionally, we configure each TRF and ViBiD with our settings: interpolation scale $\lambda$ as 1.0 and 0.5, the number of re-noising steps $k$ as 2 and 3, and the distillation step ratio $\gamma$ as 0.3 and 0.2.

### 5.2 COMPARATIVE RESULTS

As comparison methods, we choose representative time reversal sampling-based methods: TRF (Feng et al., 2024), GI (Wang et al., 2025b), FCVG (Zhu et al., 2025), and ViBiD (Yang et al., 2025a). We also include a flow-based VFI model, FILM (Reda et al., 2022), and a recent generative VFI model, DynamiCrafter (Xing et al., 2024b), for a broader comparison.

**Quantitative results.** For quantitative evaluations, we use metrics for video frame interpolation, including FID (Heusel et al., 2017), FVD (Unterthiner et al., 2019), LPIPS (Zhang et al., 2018). FID and FVD measure the distances of generated frames/videos over ground-truth sequences. LPIPS assesses perceptual similarity at frame level. We also evaluate the overall quality of the videos using VBench (Huang et al., 2024a) and VBench++ (Huang et al., 2024b). VBench provides a comprehensive assessment across multiple dimensions such as subject consistency, background consistency, aesthetic quality, image quality, motion smoothness, and temporal flickering. VBench++ typically evaluates comprehensive performance of videos with a single reference frame. Because inbetweening must treat both endpoints symmetically, we compute VBench++ for start and end frame each, then average them.

As shown in Tab. 1, our method consistently outperforms the time reversal sampling-based methods, TRF and ViBiD, across all metrics. In particular, our method achieves significant improvements in terms of FID and FVD scores, highlighting its ability to produce temporally coherent sequences with smooth motion. For VBench++, FILM gets slightly better scores than our methods in DAVIS dataset. However, this can be attributed to flow-based warping that preserves local structures near each endpoint. This comes with blurry artifacts and weaker long-range temporal consistency, which yields the lower FVD score. Overall, our method effectively addresses the issue of conflicting motion priors and achieves both fidelity and perceptual quality over SOTA methods.

---

[1]https://www.pexels.com/

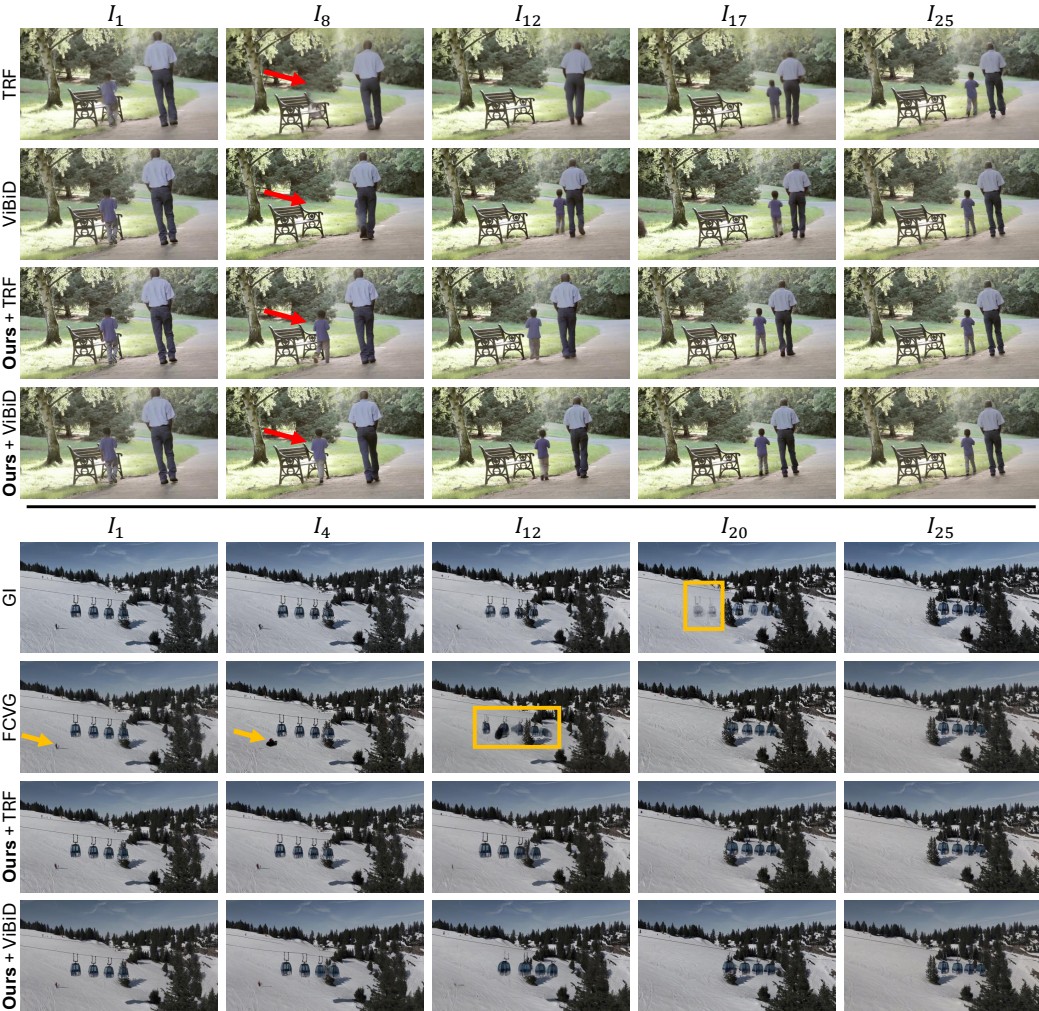

Figure 3: **Qualitative baseline comparisons.** TRF and ViBiD suffer from back-and-forth motion and intermittent disappearance, while GI and FCVG exhibit noticeable artifacts and ghosting effects. Our method yields more temporally consistent motion than the comparison methods. Additional examples are provided in the project page.

**Qualitative results.** As shown in Fig. 3, our method produces a more temporally consistent motion than the comparison methods. In the first group, TRF and ViBiD fail to preserve the child's forward-walking trajectory. Near the end frames, the child appears to walk backward or partially vanish, indicating the misalignment issue between the two paths. In the second group, GI and FCVG exhibit oscillations and ghosting artifacts. In particular, FCVG, relying on line matching, results in ambiguous artifacts, which is observed with the skier. In common, the comparison methods encounter difficulties when subjects' motion orientations are similar in both the start frame and end frame. In contrast, we validate that injecting forward motion residual into the backward path is enough to represent desirable object motions with fewer artifacts.

**User study.** To further evaluate human preference beyond quantitative metrics, we conduct a comprehensive user study via Amazon Mechanical Turk (Crowston, 2012) following prior works (Shin et al., 2024; 2025). Each participant is presented with pairs of the start and end frames, followed by randomly 8 candidate videos generated by different methods. To avoid ordering bias, the display order is randomized for every sequence.

Our user study is designed with three types of questionnaires: (1) *Ranking*: participants are asked to rank videos in order of overall naturalness and temporal coherence, focusing on how plausibly the generated sequence links the start and end frames. These rankings are converted into scores in reciprocal order, ranging from 3.5 to -3.5. (2) *Artifact detection*: participants are asked to select all videos

Table 2: **Comparison results of user study.** Best results are **bold**, and second-best are underlined.

| Method | Alignment ↑ | Artifact ↓ | Unrealistic ↓ | Method | Alignment ↑ | Artifact ↓ | Unrealistic ↓ |
|---|---|---|---|---|---|---|---|
| FILM | - 0.4060 | 62.74% | 54.76% | FCVG | 0.0988 | 20.36% | 19.17% |
| DynamiCrafter | 0.0190 | 34.64% | 37.14% | ViBiD | - 0.0678 | 28.10% | 25.24% |
| TRF | - 0.3119 | 28.09% | 25.24% | **Ours** + TRF | **0.3060** | 20.36% | 22.62% |
| GI | 0.1179 | 22.26% | 13.57% | **Ours** + ViBiD | 0.2440 | **8.93%** | **9.88%** |

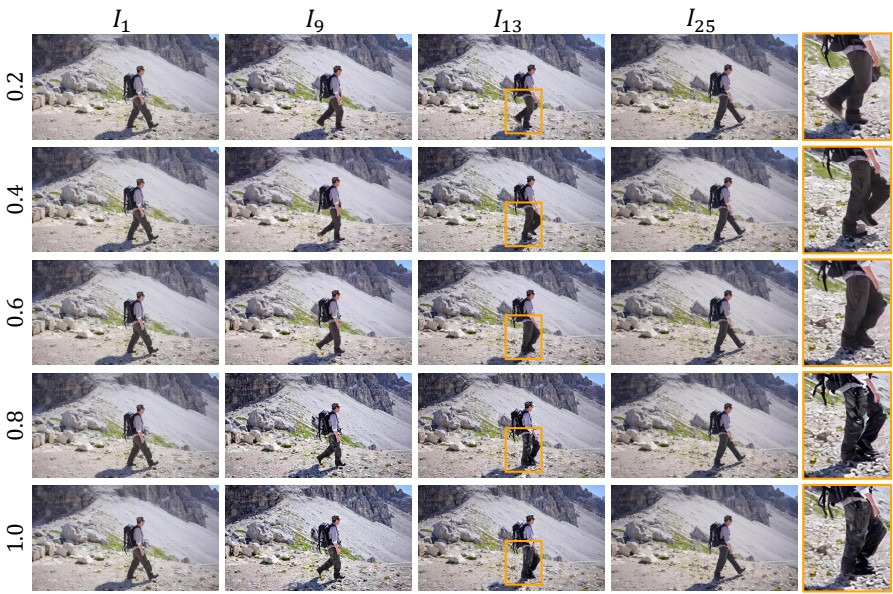

Figure 4: **Ablation study on the effect of distillation ratio** $\gamma$. We vary the distillation step ratio $\gamma \in \{0.2, 0.4, 0.6, 0.8, 1.0\}$, where $\gamma = 0.2$ corresponds to the default setting and $\gamma = 1$ applies our method at every denoising step.

that exhibit noticeable visual artifacts, such as distortions, ghosting, or inconsistent textures. (3) *Unrealistic motion identification*: participants are asked to choose all videos that contain unrealistic or physically implausible movements, which are closely related to perceptual plausibility.

We collect responses from a total of 30 participants across 28 randomly sampled video groups to ensure the statistical reliability of the study. As shown in Tab. 2, our method achieves the highest preference in the ranking task, while being selected least frequently in both the artifact and unrealistic motion categories. These results demonstrate that our approach outperforms competitive baselines in terms of perceptual plausibility and human preference, providing strong evidence of its effectiveness in practical scenarios.

## 5.3 ABLATION STUDIES

We conduct ablation studies on DAVIS dataset to evaluate the impact of distillation step ratio $\gamma$, re-noising steps $k$, and interpolation scale $\lambda$. The results are summarized in Table 3.

Within the parallel approach, LPIPS and FID are minimized at $\gamma = 0.3$ and $k = 2$, whereas FVD prefers weaker distillation and fewer re-noising steps. This is because TRF fuses the two conditional paths at every step, and the opposite motion prior is continually re-introduced after MPD, making the process more sensitive. Averaging the two paths partially cancels out the conflict and improves temporal coherence, while stronger MPD process can favor the frame-level fidelity at the cost of temporal smoothness.

For the sequential time reversal sampling, we observe a clear and consistent optimum at $\gamma = 0.2$, $k = 3$, and $\lambda = 1.0$, achieving the best scores. This indicates that strong early single-prior distillation with no interpolation is beneficial for the sequential method. Once the backward path is aligned to the forward motion prior in the early phase, subsequent steps rarely introduce conflicting priors, so increasing $k$ steadily helps to improve the temporal and perceptual quality of videos.

Table 3: **Ablation results** for distillation steps ratio $\gamma$, re-noising steps $k$, and interpolation scale $\lambda$. (a)–(c) correspond to **Ours** + TRF, and (d)–(f) correspond to **Ours** + ViBiD. Best results are **bold**.

| (a) Distillation step ratio ($\gamma$) | | | | (b) Re-noising steps ($k$) | | | | (c) Interpolation scale ($\lambda$) | | | |
|---|---|---|---|---|---|---|---|---|---|---|---|
| $\gamma$ | LPIPS $\downarrow$ | FID $\downarrow$ | FVD $\downarrow$ | $k$ | LPIPS $\downarrow$ | FID $\downarrow$ | FVD $\downarrow$ | $\lambda$ | LPIPS $\downarrow$ | FID $\downarrow$ | FVD $\downarrow$ |
| 0.3 | **0.2212** | **34.910** | 612.17 | 1 | 0.2246 | 35.149 | **588.27** | 0.5 | **0.2212** | **34.910** | **612.17** |
| 0.2 | 0.2238 | 35.408 | 576.01 | 2 | **0.2212** | **34.910** | 612.17 | 1.0 | 0.2264 | 35.612 | 654.72 |
| 0.1 | 0.2236 | 35.268 | **573.13** | 3 | 0.2248 | 35.765 | 662.75 | - | - | - | - |

| (d) Distillation step ratio ($\gamma$) | | | | (e) Re-noising steps ($k$) | | | | (f) Interpolation scale ($\lambda$) | | | |
|---|---|---|---|---|---|---|---|---|---|---|---|
| $\gamma$ | LPIPS $\downarrow$ | FID $\downarrow$ | FVD $\downarrow$ | $k$ | LPIPS $\downarrow$ | FID $\downarrow$ | FVD $\downarrow$ | $\lambda$ | LPIPS $\downarrow$ | FID $\downarrow$ | FVD $\downarrow$ |
| 0.3 | 0.2421 | 39.855 | 574.05 | 1 | 0.2379 | 39.961 | 568.06 | 0.5 | 0.2242 | 37.837 | 539.99 |
| 0.2 | **0.2220** | **37.241** | **527.05** | 2 | 0.2341 | 39.368 | 545.25 | 1.0 | **0.2220** | **37.241** | **527.05** |
| 0.1 | 0.2374 | 39.949 | 545.25 | 3 | **0.2220** | **37.241** | **527.05** | - | - | - | - |

## 5.4 THE EFFECT OF DISTILLATION RATIO

We conduct another ablation study on the variation of the distillation step ratio $\gamma$ up to 1.0. As shown in Tab. 4, increasing $\gamma$ consistently leads to worse scores across all metrics. The cropped examples in Fig. 4 further show that applying our method with $\gamma > 0.3$ does not further improve motion consistency, but rather introduces undesirable pixel-level biases and degrades visual fidelity. Both quantitative and qualitative studies support our choice to apply our method in the early stage of sampling, rather than throughout the entire denoising process.

Table 4: **Effect of the distillation ratio** $\gamma$. Best results are **bold**.

| $\gamma$ | LPIPS $\downarrow$ | FID $\downarrow$ | FVD $\downarrow$ |
|---|---|---|---|
| 0.2 | **0.2220** | **37.241** | **527.05** |
| 0.4 | 0.2478 | 45.636 | 634.11 |
| 0.6 | 0.2562 | 55.873 | 813.08 |
| 0.8 | 0.2679 | 68.007 | 973.64 |
| 1.0 | 0.2721 | 75.544 | 1086.6 |

## 5.5 COMPUTATIONAL EFFICIENCY

We compare computational cost with other I2V diffusion based methods, as summarized in Tab. 5. DynamiCrafter requires additional training on a I2V diffusion model for the generative inbetweening task. Likewise, GI and FCVG rely on fine-tuning SVD models, which also require substantial computational resources. During inference, our method denoises the temporally forward path and then adds a few extra re-noising steps to make the two paths align. Due to these steps, the inference time can be slightly longer than FCVG and ViBiD. However, this small extra cost yields better alignment and fewer artifacts in generated videos.

Table 5: **Comparisons on computational efficiency.**

| Method | Train | Inference time (s) | VRAM Usage (GB) | Resolution |
|---|---|---|---|---|
| DynamiCrafter | ✓ | 26 | 11.2 | $16 \times 512 \times 320$ |
| TRF | ✗ | 429 | 13.6 | $25 \times 1024 \times 576$ |
| GI | ✓ | 663 | 23.4 | $25 \times 1024 \times 576$ |
| FCVG | ✓ | 134 | 24.1 | $25 \times 1024 \times 576$ |
| ViBiD | ✗ | 108 | 19.2 | $25 \times 1024 \times 576$ |
| **Ours** + TRF | ✗ | 143 | 19.2 | $25 \times 1024 \times 576$ |
| **Ours** + ViBiD | ✗ | 141 | 19.2 | $25 \times 1024 \times 576$ |

## 6 CONCLUSION

In this work, we analyze the bidirectional path misalignment problem in existing time reversal sampling through the lens of optimization problem. Based on the analysis, we present **Motion Prior Distillation (MPD)**, a training-free sampling method that resolves motion prior conflict in existing time reversal samplers to enhance generative inbetweening task. MPD replaces two conflicting temporal priors with a single coherent motion prior from the start frame, and distills it through the backward denoising path, yielding a coherent trajectory that satisfies both endpoint constraints. By integrating MPD into time reversal sampling methods, we demonstrate that MPD synergistically reduces temporal discontinuities and visual artifacts, and achieves more appealing results both quantitatively and qualitatively than SOTA methods.

ACKNOWLEDGMENTS

This work was supported by the Institute of Information & Communications Technology Planning & Evaluation (IITP) grants funded by the Korea government (MSIT) (RS-2020-II201361 and RS-2025-25441838), by the National Research Foundation of Korea (NRF) grant funded by the Korea government (MSIT) (RS-2024-00338439), and by the Yonsei University Future-Leading Research Initiative of 2025 (2025-22-0409).

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

# A    ANALYSIS ON DENOISED ESTIMATES

Figure I: **Analysis on denoised estimates.** At the midpoint of time reversal sampling, we take the forward and backward denoised estimate, align the latter to the temporal order, and inspect their difference.

For a clearer understanding of our motivation in Sec. 4.1, we conduct an additional experiment in which the forward and backward denoised estimates, $\hat{x}_{0,c_{\text{start}}}$ and $\hat{x}'_{0,c_{\text{end}}}$, are obtained at an intermediate time step ($t = 0.5T$), both with and without applying our method. Note that the backward denoised estimate is temporally flipped to enable direct comparison. As shown in the first two rows of Fig. I, existing methods produce intermediate frames with implausible motion. The forward and backward trajectories attempt to reflect two incompatible motion priors simultaneously, revealing a clear motion conflict. In contrast, as shown in the last two rows of Fig. I, our method maintains a consistent motion trajectory in both temporal paths, generating coherent intermediate frames without such conflicts.

# B    FINE-TUNING METHODS WITH TIME REVERSAL SAMPLING

In this section, we deeply discuss the fine-tuning methods that incorporate the time reversal sampling strategy. Existing fine-tuning approaches (Wang et al., 2025b; Zhu et al., 2025) share two major limitations. First, they still rely on incompatible motion priors at inference time, so mismatches between forward and backward trajectories are not fundamentally resolved. Second, both methods require additional fine-tuning, which demands substantial computational cost. In contrast, our method aligns the two temporal paths without additional training, and integrates into existing time reversal samplers with only a minor change to the sampling loop.

GI (Wang et al., 2025b) improves backward motion fidelity by fine-tuning SVD through rotated temporal self-attention maps. While this design encourages consistency between two paths, the two paths are still driven by different motion priors. Additionally, the backward-motion network is trained only on a small collection of videos, which may not fully capture the diverse backward motion patterns. As shown in Fig. VI, this can lead to motion conflicts such as two surfers being generated, even though there must be one. Instead, our method reconstructs a backward estimate directly from the forward motion residuals, so that the backward path no longer introduces its own motion prior but instead follows the time-reversed motion induced by the start frame, leveraging the faithful forward motion prior of SVD.

FCVG (Zhu et al., 2025) adopts frame-wise conditions by extracting matched line segments from the two keyframes. Then, they interpolates the frame-wise correspondences over time, and then feeds them into SVD as the guidance. This works well when scenes contain strong, well-defined structural edges, but it has practical limitations. The method is highly sensitive to the quality of the line extraction and matching pipeline. Thus, failures in detection or matching directly lead to unstable or distorted interpolations. As shown in Fig. IV, such failures appear as noticeable ghosting artifacts on the carrier.

## C    EXPERIMENTS ON LARGE TEMPORAL GAPS

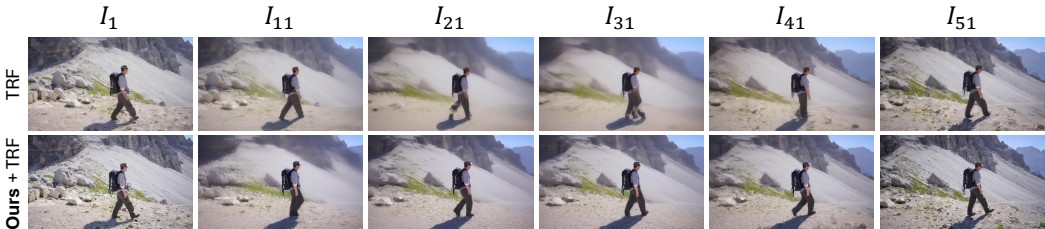

Figure II: **Qualitative results over a 50-frame gap.**

We conduct additional experiments on large temporal gaps, where the two keyframes are separated by 50 frames. In this challenging setting, both the previous time reversal sampling methods and our method exhibit noticeable degradation in generated videos. However, our method still reduces severe ghosting and back-and-forth motion compared to the baselines, thereby producing more plausible intermediate frames. As shown in Fig. II, TRF often yields duplicated or intermittent legs, whereas incorporating our method with TRF generates more coherent inbetweening results.

## D    DISCUSSIONS ON EFFECTIVE AND CHALLENGING SCENARIOS

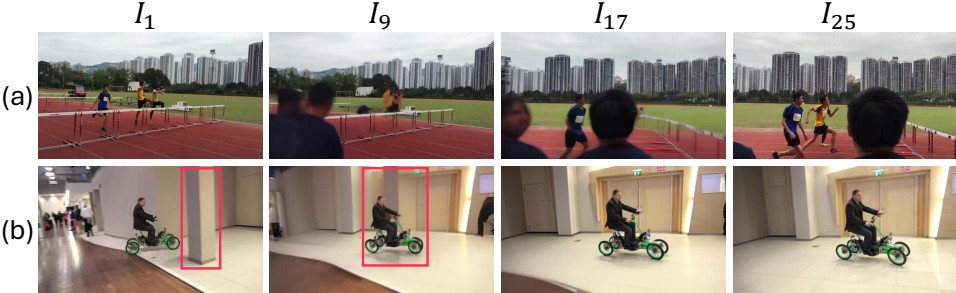

Figure III: **Failure cases.** Our method struggles when (a) the end keyframe introduces entirely new objects or (b) the scene undergoes large-scale rearrangements between two keyframes.

One of the key components of our method is the endpoint initialization in Eq. (15) that initializes the backward noise $\epsilon_{\text{bwd}}$ using the latent encoding of the end frame $z_{\text{end}}$. This implicitly assumes that the end frame of the forward path is reasonably consistent with the ground truth end frame. When the forward trajectory ends far from the ground truth end frame, Eq. (15) may become a less informative initialization. In such cases, the shared motion prior can be biased toward an inaccurate end frame, so smalls residual discrepancies in object placement or appearance may remain.

We experimentally find that our method is most effective when both keyframes contain the same object and share a semantically coherent motion as presented in Appendix E. In such scenarios, the forward motion prior provides the reliable global trajectories, and distilling it into the backward path successfully reduces undesirable artifacts. However, as discussed in previous works (Wang et al., 2025b; Zhu et al., 2025), MPD still struggles with the inevitable problems that arise when the end frame introduces entirely new objects or undergoes large-scale scene rearrangements. Representative failure cases are presented in Fig. III.

To better handle these challenging scenarios, we are currently exploring extensions of our approach on large-scale I2V diffusion models such as CogVideoX (Yang et al., 2025b), which adopt a DiT-based architecture (Peebles & Xie, 2023) and can generate over 80 frames.

# E    ADDITIONAL QUALITATIVE RESULTS

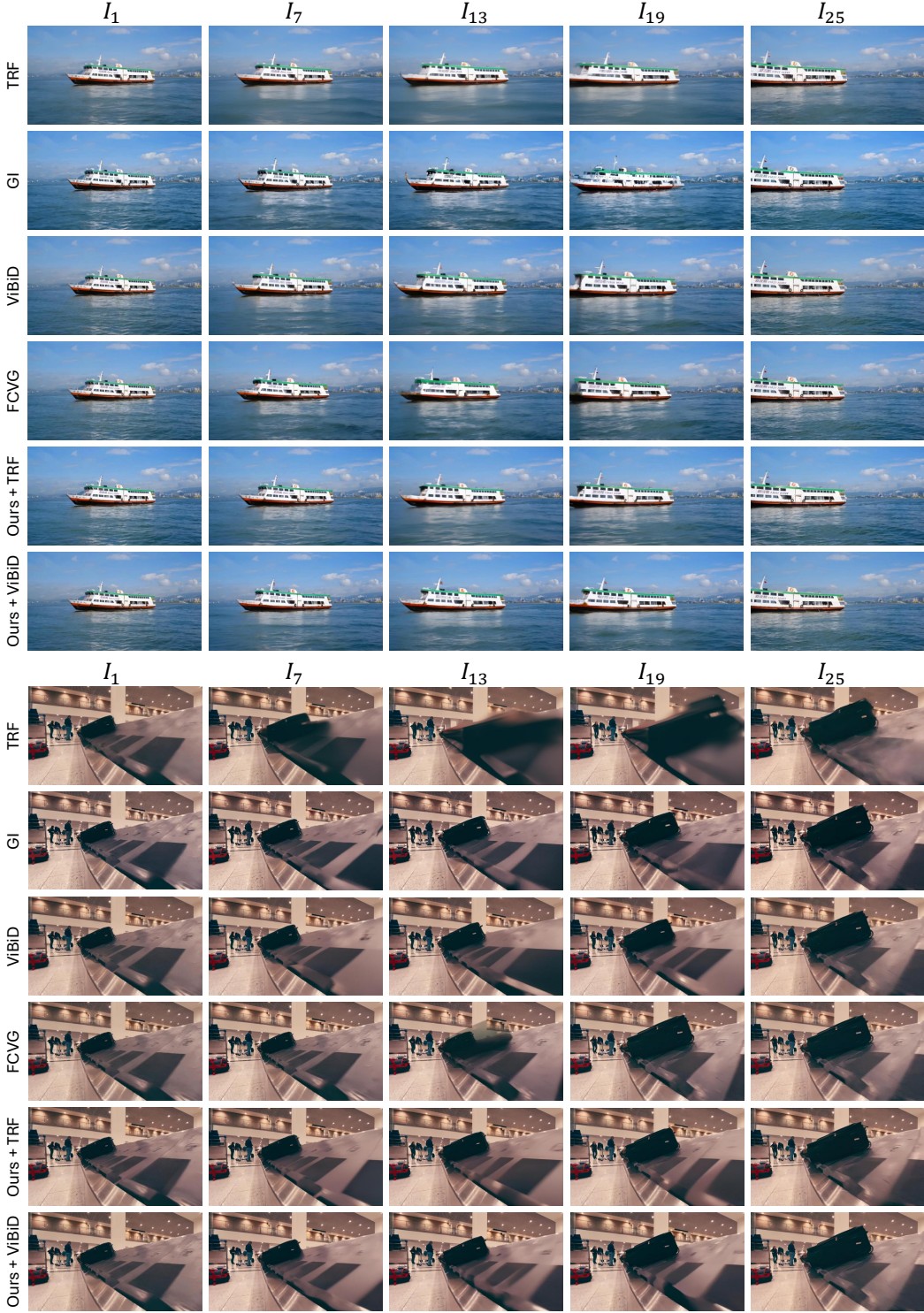

Figure IV: Additional comparison results with baseline models.

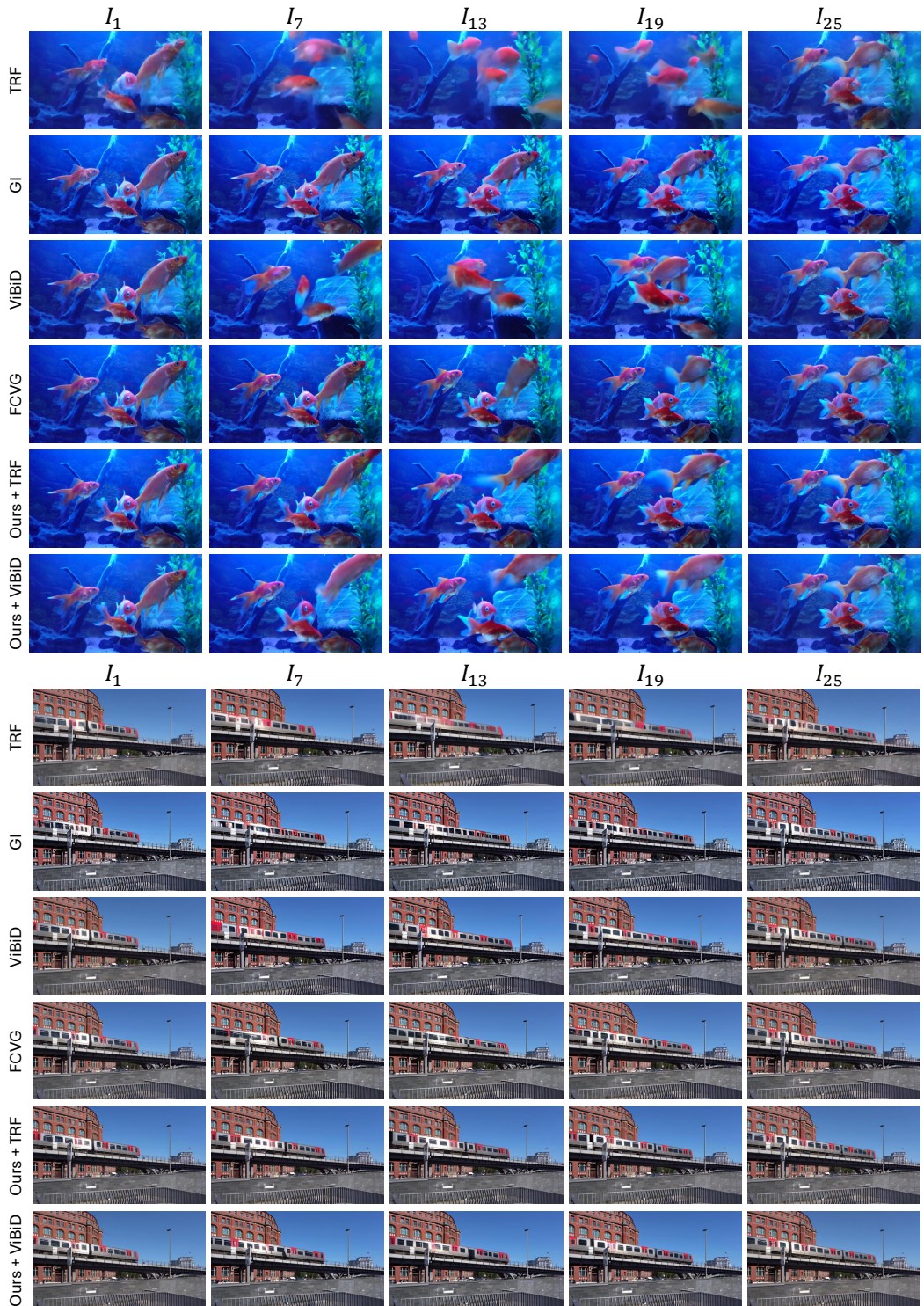

Figure V: Additional comparison results with baseline models

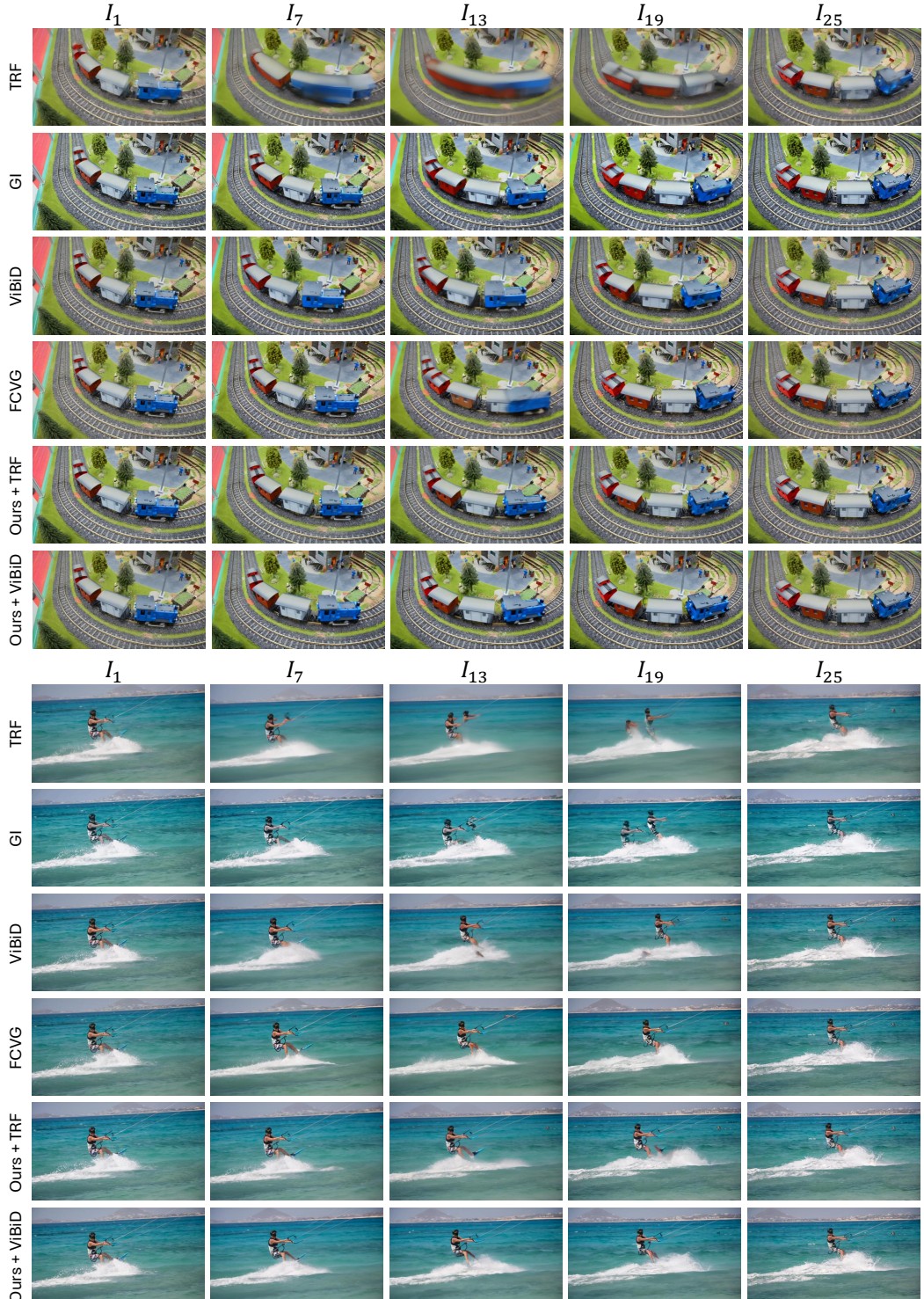

Figure VI: Additional comparison results with baseline models

