# OpenReview forum: "Motion Prior Distillation in Time Reversal Sampling for Generative Inbetweening"
_ICLR.cc/2026/Conference — ICLR 2026 Poster_

### Official Review · Reviewer_6htp · 2025-10-27

**Soundness:** 4
**Presentation:** 4
**Contribution:** 3
**Rating:** 6
**Confidence:** 5

**Summary:**

Inference-time sampling strategies for video interpolation, which leverage the generative priors of large-scale pretrained image-to-video (I2V) models without additional training (e.g., TRF and ViBiDSampler), often suffer from temporal discontinuities. These methods typically fuse or alternate between temporally forward paths (starting from latent $z_{start}$) and backward paths (starting from latent $z_{end}$). However, misalignment between the two paths, arising from differently conditioned motion priors, frequently leads to inconsistent temporal coherence.

This work proposes an inference-time distillation technique that transfers motion priors from the forward path (starting from $z_{start}$) to the backward path during the early sampling stage ($T \rightarrow (1 - \gamma)T$). Specifically, the method computes frame-wise residuals from the forward path and recursively injects these residuals into the end-frame latent $z_{end}$ to generate the backward path. As a result, the backward path is effectively constrained with $z_{end}$, while the diffusion prior is applied only to the forward path—thereby reducing misalignment and improving temporal coherence.

After applying inference-time distillation in the early sampling stage ($T \rightarrow (1 - \gamma)T$), the method transitions to existing sampling frameworks such as TRF and ViBiDSampler, demonstrating that the proposed technique effectively enhances their performance.

**Strengths:**

1) Good motivation and novel framework

Motivated by the goal of resolving the bidirectional path misalignment problem, the authors propose a novel framework called Motion Prior Distillation. This approach injects motion information from the forward path into the end-frame latent $z_{end}$ to guide the generation of the backward path. As a result, the method constrains to terminate at the end-frame latent $z_{end}$ while utilizing only the forward-path motion prior, effectively mitigating bidirectional path misalignment. Using only the forward-path motion prior in this context is both novel and insightful.

2) Comprehensive experiments

The authors clearly demonstrate that applying Motion Prior Distillation during the early sampling stage enhances the performance of existing inference-time frameworks such as TRF and ViBiDSampler. In addition, they conduct extensive ablation studies to explore the underlying principles of the proposed approach, which are appropriately designed and well-analyzed in terms of quantitative metrics.

3) Zero-shot improvement

It is worth highlighting that the proposed method improves upon prior works while maintaining a zero-shot setting. This makes the contribution particularly relevant and appealing to researchers working on video diffusion and zero-shot video interpolation.

**Weaknesses:**

1) Limited explanation of early-stage application

After carefully reviewing the submission, it remains unclear why the proposed method should be applied only during the early stage of the sampling process. The authors briefly attribute this choice to the “coarse-to-fine property of diffusion sampling,” but a more detailed explanation is needed. In particular, the paper should clarify why the proposed method is especially effective for correcting global or low-frequency structures, thereby helping readers understand the underlying principle of why the approach works well in the early stage.

2) Additional ablation study on distillation step ratio

Building on the previous point, an ablation study investigating different distillation step ratios, especially values larger than 0.3, would help substantiate the authors’ claims regarding the optimal range and behavior of the proposed method.

3) Limited demonstration of computational efficiency

While prior works typically report quantitative efficiency metrics such as runtime and VRAM usage, this paper lacks such evaluations. Including these measurements would make the efficiency claims more convincing and allow for fair comparison with existing approaches.

**Questions:**

1) Clarification on the necessity of switching methods

As mentioned in Weakness 1, it is unclear why the proposed method cannot perform the full sampling process independently. The authors should clarify why it is necessary to switch to another sampling framework midway, rather than continuing the proposed method throughout the entire process.

2) Additional ablation on distillation step ratio

It would be helpful to include additional ablation studies that extend the distillation step ratio up to 1.0. This would make it clearer how the proposed method behaves when applied across the full sampling range and help justify the chosen configuration.

3) Quantitative efficiency metrics

Please report quantitative efficiency metrics such as runtime (s/frame) and VRAM usage (GB) to support the claimed efficiency and enable fair comparison with prior methods.

---

> ### Author Response · Authors · 2025-11-20
>
> We sincerely appreciate the reviewer’s constructive feedback and their recognition of our strengths. The corresponding revisions in the paper are highlighted in orange-colored texts.
>
> > **W1, Q1**: Limited explanation of early-stage application. Clarification on the necessity of switching methods.
>
> **A**: Thank you for raising this important question. Our design choice is grounded in the coarse-to-fine nature of diffusion sampling: prior works [1,2,3] have shown that early steps at high noise levels primarily determine global structure and motion, while later steps mainly refine high-frequency details and appearance. Complementary studies [4,5,6] further demonstrate that focusing guidance on these early steps yields better visual quality. Thus, modifying the motion prior is most effective when the trajectory is still being shaped globally. Once the global motion is established, further disturbing the denoising dynamics tends to harm fine details rather than improve alignment.
>
> In the revised paper, we clarify this intuition in the method section and support it empirically through additional ablation study in Sec. 5.4.
>
> **References**
>
> [1] Rissanen et al. “Generative modelling with inverse heat dissipation.” *ICLR* (2023).
>
> [2] Kim et al. “ Leveraging early-stage robustness in diffusion models for efficient and high-quality image synthesis.” *NeurIPS* (2023).
>
> [3] Wu et al. “Importance-based token merging for efficient image and video generation.” *ICCV* (2025).
>
> [4] Lee et al. “Beta sampling is all you need: Efficient image generation strategy for diffusion models using stepwise spectral analysis.” *WACV* (2025).
>
> [5] Lee et al “Videoguide: Improving video diffusion models without training through a teacher’s guide.” *CVPR* (2025).
>
> [6] Park et al. “Inference-time diffusion model distillation.” *ICCV* (2025).
>
> > **W1, Q2**: Additional ablation study on distillation step ratio.
>
> **A**: Thank you for this constructive suggestion, which helps clarify the effect of applying our method in the early denoising stage. Following your comment, we conducted an additional ablation study that varies the distillation ratio up to 1.
>
> | γ | LPIPS | FID | FVD |
> |---|---|---|---|
> | **0.2** | **0.2220** | **37.241** | **527.05** |
> | 0.4 | 0.2478 | 45.636 | 634.11 |
> | 0.6 | 0.2562 | 55.873 | 813.08 |
> | 0.8 | 0.2679 | 68.007 | 973.64 |
> | 1.0 | 0.2721 | 75.544 | 1086.6 |
>
> We revise the paper containing the table to Tab. 4 and qualitative results to Fig. 4. As shown in the table, increasing distillation ratio consistently leads to worse scores across all quantitative metrics. The cropped examples in Fig. 4 further show that applying our method during the later steps does not further improve motion consistency. Instead, it introduces undesirable pixel-level biases and degrades visual fidelity. Both the quantitative and qualitative studies support our choice to apply our method in the early denoising stage, rather than throughout the entire denoising process. We added the results in Sec. 5.4 of the revised paper.
>
> > **W3, Q3**: Limited demonstration of computational efficiency.
>
> **A**: We appreciate the reviewer’s suggestion to provide a clearer comparison of computational efficiency. In the revised paper, we report quantitative efficiency metrics for our method and other I2V diffusion based methods in Appendix C.
>
> | Method | Train | Inference Time (s) | VRAM (GB) | Resolution |
> |---|---|---|---|---|
> | DynamiCrafter | ✓ | 26 | 11.2 | 16×512×320 |
> | TRF | ✗ | 429 | 13.6 | 25×1024×576 |
> | GI | ✓ | 663 | 23.4 | 25×1024×576 |
> | FCVG | ✓ | 134 | 24.1 | 25×1024×576 |
> | ViBiD | ✗ | 108 | 19.2 | 25×1024×576 |
> | **Ours**+TRF | **✗** | **143** | **19.2** | **25×1024×576** |
> | **Ours**+ViBiD | **✗** | **141** | **19.2** | **25×1024×576** |
>
> DynamiCrafter requires additional training on an I2V diffusion model for the generative inbetweening task. Likewise, GI and FCVG rely on fine-tuning SVD models, which also require substantial computational resources. During inference, our method denoises the temporally forward path and then adds a few extra re-noising steps to make the two paths align. Due to these steps, the inference time can be slightly longer than FCVG and ViBiD. However, this small extra cost yields better alignment and fewer artifacts in generated videos.

---

### Official Review · Reviewer_mj1L · 2025-10-29

**Soundness:** 2
**Presentation:** 2
**Contribution:** 2
**Rating:** 4
**Confidence:** 3

**Summary:**

This paper aims to solve the task of generative inbetweening, using the well-established image-to-video diffusion models. This paper tries to solve a problem of motion conflict when using time reversal sampling, which is the misalignment of motion paths estimated from two different directions of predictions - forward and backward. Existing methods try to fuse the estimations of both directions, and did not focus much on fixing possible misalignment issues. The authors propose Motion Prior Distillation, which use the residual noise predictions estimated from the forward path and distill them to the backward path in reverse, to match the predicted paths of both directions.

**Strengths:**

- The problem definition is clear.
- The results of the proposed method seem to be strong, outperforming existing work.

**Weaknesses:**

1. According to my understanding, in short, the proposed method aims to model the noise residual between frames, especially from the forward path, and distill it to the backward path for alignment. However, I feel quite unsure how this could well-align the motions. Figure 1 (c) seems to be a good description, and this also display my concern. In Fig.1(c), the position of the red car in the forward path and the backward path differ. To be more specific, in the backward path, starting from the end (green car), using the residual distilled from the forward path ends up in a different location of the red car, which is different from the reference start frame (red car). To my understanding, I believe that it is possible to cause a misalignment, and I wonder how this could rather resolve misalignment and motion conflict. Fusing the estimates of two paths may be a part which handles this part, but in that case, I am not sure of how the proposed method could theoretically contribute to the problem of resolving motion conflicts. Could the authors provide further explanation and clarify on this?
2. Theoretically speaking, I understood the problem definition described by the authors in Sections 3 and 4.1. However, I am not following whether this is a “motion” “conflict”. For instance, Fig.1(d) is a good example to support my point. To my perspective, the forward path and the backward path both seem to locate the cars at similar positions, aligned quite well, despite blurry and unsatisfactory quality. Despite being blurry, the position of the visual artifacts, which is presumably the location where the model tried to draw the car, seems to be aligned well. Thus I find this problem to be more of a failure of inserting the object (car), rather than failure of modeling motions. I think it is not a problem with “motion”, but a failure of object injection / disappearance; and think the motions of forward and backward paths are not in “conflict”, with good alignment. It could be a matter of how we define things, but currently, I think the word choice of definition is a little misleading. Could the authors provide some discussions on this?

The weaknesses mentioned above are more of a question / discussion, rather than flaws. With these questions above clearly resolved, I am open to raise the rating.

**Questions:**

- In lines 73-74, could the authors provide a little more explanation on how connecting the forward and backward paths differ from aligning them? Connection of forward and backward paths would inevitably require alignment, and I feel unsure of what this precisely means.

---

> ### Author Response · Authors · 2025-11-20
>
> We sincerely thank the reviewer for the positive assessment of the problem setting and results, and for the insightful questions. Below we have tried to address each point in detail and clarify how our method resolves motion conflicts. The corresponding revisions in the paper are highlighted in green-colored texts.
>
> > **W1**: Does distilling forward residuals into the backward path really “align” motions?
>
> **A**: First of all, we kindly request the reviewer to check the revised version of the paper. We appreciate this question and agree that Fig. 1 (c) shows a small residual offset between the distilled red car and the ground-truth position, as intended. Our goal is not to reconstruct the start frame exactly from the end frame via residuals, but to enforce a shared motion prior between the forward and backward directions. Existing methods maintain two independent motion priors: one induced by the start frame and another by the end frame, as displayed in Fig. 1 (d) and (e). When these priors strongly disagree on the trajectory (e.g., route and destination of the car), simply connecting their outputs inevitably produces ghosting or reverse play as shown in Fig. 1 (f).
>
> In contrast, our method removes the end-conditioned motion prior in the early denoising stage and reconstructs the backward path from the forward motion residuals. The relaxed objective in Eqs. (20)–(21) then minimizes the discrepancy between two paths that now share a single prior, rather than between two conflicting priors as in Eq. (11). From this perspective, our method reshapes the optimization problem itself. Instead of trying to reconcile two incompatible trajectories as shown in Fig. 1 (d) and (e), we align both directions under one motion prior and only then apply time reversal sampling. Furthermore, with additional renoising steps, we gradually steer the trajectory toward a coherent path consistent with the shared motion prior.
>
> > **W2**: Is this really a “motion conflict” and not just object insertion failure?
>
> **A**: In our paper, we use the term motion conflict to describe a trajectory-level mismatch between the forward and backward motion priors over the entire sequence, rather than a discrepancy at a single frame. To be specific, the forward and backward paths each induce a different motion about where the object should be and how it should move as shown in Fig. 1 (d) and (e). When these two priors disagree and are later combined by time reversal sampling, the disagreement manifests as ghosting, back-and-forth motion, or intermittent disappearance.
>
> We agree that in the original submission, Fig. 1 (d) may have given the impression that the car locations are roughly aligned. In the revised version, we thus (i) increase the number of displayed intermediate frames of the result from 4 to 6 in Fig. 1 (f), and (ii) move the intermediate timestep visualizations to Appendix A with a clearer analysis. In Fig. 1 (f), the car exhibits the reverse play and ghosting at the intermediate frames, indicating that the two directions do not agree on the progression of the motion.
>
> In Appendix A, we visualize the forward and backward denoised estimates at the midpoint of sampling in Fig. I. As shown in the first two rows of Fig. I, we can observe that the two trajectories attempt to reflect two incompatible motion priors without truly aligning them. This is a direct consequence of existing time reversal sampling strategies. They simply connect the forward and backward paths (by fusion or alternation), but never enforce a consistent shared motion prior between them.
>
> > **Q1**: Lack of explanation on lines 73-74.
>
> **A**: In the revised paper, we clarify the distinction between connecting and aligning, and additionally illustrate this difference in Fig. 2 (a) and (b).
>
> As illustrated by the green arrows in Fig. 2 (a) and (b), existing methods connect the two independently generated paths by either (a) linearly fusing the forward and backward denoised samples at each step, or (b) alternately denoising each path. Crucially, both methods do not modify the underlying motion priors induced by the start- and end-conditioned paths. As shown in Fig. 1 (d)-(f), the forward and backward motion priors are used simultaneously but remain inconsistent throughout sampling, so this **“connection”** simply merges two incompatible trajectories and often leads to ghosting, back-and-forth motion, or reverse play.
>
> In this sense, simply connecting two paths does not guarantee alignment of their motion priors; it only combines whatever each prior proposes. Our method first aligns the motion priors by our proposed distillation, thereby mitigating such motion conflicts. These clarifications and the updated explanation are highlighted in green in the revised paper.

---

> > ### Comment · Reviewer_mj1L · 2025-11-27
> >
> > Thank you for the clarifications and the update in the manuscript.
> > I figured that I had some misunderstandings on the problem definition, and it has been clarified by the authors' rebuttal.
> > I think the problem definition is interesting and well-motivated, and have raised my score.
> >
> > Despite the clarification, I would like to suggest a change on the term on resolving "motion conflict" to something like resolving "motion prior bias" or "forward-generation bias". I think this choice of terminology was quite misleading for me. Although it could be considered a "conflict" in motions, but I think the main motivation of this paper is on resolving the motion priors of I2V models, which are "biased" towards forward-generation. The updated explanations and figures did help, but was not easy at first glance. In the current form, some parts still sound like the "conflict-resolvement" is on aligning discrepancies of two predictions at one frame. This is simply a suggestion, and would like to hear the authors' thoughts on this. If there is still something that I am missing, please point me out.
> >
> > In addition, I have one more question on the proposed method. This has risen after clarification on the problem definition. I wonder if motion prior distillation does not suffer from object misalignments. Overall, I think the proposed method could work, but in some aspect, I wonder if there is no problematic scenarios. For instance, the last noise residual of last frame pair from the forward path could be misaligned with the first frame pair of the backward path. In this case, the noise residual would be computed based on the locations of objects, background etc., but if it is applied to a frame with objects / background with different locations, wouldn't it be problematic?
> > Using optical flows as an analogy, I am concerned of this type of misalignment:
> > let an optical flow map $F_{t\rightarrow t+1}$ be optical flow map from frame $I_t$ to $I_{t+1}$, and in this case $F_{t\rightarrow t+1}$ should be aligned with $I_t$ and $F_{t\rightarrow t+1}$ would be the flow to apply any operations (e.g. Forward Warping). Applying $F_{t+1\rightarrow t}$ to $I_t$ could work in some cases due to vast overlap between $I_t$ and $I_{t+1}$, but the object alignment itself is flawed, and is prone to errors. Viewing the residuals as sort of an optical flow, which contains pixel-level fine details of objects, etc., I think there could be such misalignment issues. Could the authors provide little more insights on this matter?

---

> ### Author Response · Authors · 2025-11-28
>
> We sincerely thank the reviewer for the constructive feedback and for raising the score. We are glad that our previous response successfully clarified the problem definition and resolved the initial concerns.
>
> We also appreciate the insightful suggestion regarding the terminology. We agree that terms like "motion prior bias" or "forward-generation bias" are more precise than "motion conflict." We will reflect this change in the final manuscript.
>
> Regarding your concern about potential object misalignment in motion prior distillation, we are currently finalizing our response to these points and will update you shortly.

---

> ### Author Response · Authors · 2025-12-01
>
> > ***On terminology of "motion conflict" vs. "motion prior bias"***
>
> We appreciate the reviewer’s suggestion to replace the term “motion conflict” with terminology that more explicitly reflects the underlying cause, such as “motion prior bias” or “forward-generation bias.” In our work, we use the term *motion conflict* to denote the **visible discrepancy** between the motions induced by the start- and end-conditioned sampling paths, rather than the underlying cause itself. In Sec. 1, we explain these motion conflicts as a consequence of the **forward-generation bias** of existing I2V models, which are trained to predict forward consecutive frames and therefore tend to produce forward-looking sequences. Overall, we agree with the reviewer that the term “motion conflict” can be misleading, as it may give the impression that our main contribution is merely to reconcile two predictions at a single frame instead of addressing the biased motion prior. In the revised manuscript, we therefore (i) explicitly define *motion conflict* as the visible discrepancy between the two temporal paths, and (ii) revise the problem statement to emphasize the **forward-generation bias** as the central phenomenon that our method aims to address in Sec. 1. We believe these clarifications make the motivation and scope of our work clearer.
>
> > ***On potential object misalignment when distilling motion prior***
>
> We also appreciate the reviewer’s insightful question on whether our method can suffer from object misalignment, and the analogy with optical flow maps. We would like to clarify how our method is applied and how it behaves differently from optical flow-based operations.
>
> First, unlike an optical flow field, which is a pixel-wise displacement map defined on the coordinates of a specific frame, our motion prior residual is a direction in latent/noise space. It does not prescribe a one-to-one mapping between pixels of two frames. Instead, it behaves more like a coarse, trajectory-level gradient that nudges the noisy latent along a global motion direction. This is a key difference from optical flow. Optical flow must be strictly aligned with a specific reference frame, while our residual is defined in a shared latent space and is used as a soft guidance signal.
>
> Second, we apply MPD only at early, high-noise denoising steps, where the latent primarily represents global structure and motion rather than fine pixel-level details. In these steps, we additionally introduce a small number of re-noising steps to steer the trajectory toward the time-reversed forward motion prior. After these early stages, we disable our distillation process and switch back to the original time reversal sampling for the later, low-noise steps that refine local textures and pixel-level alignment. This design reflects our intention that MPD should shape the coarse trajectory rather than force pixel-level correspondences in the final frames.
>
> Importantly, we note that the reviewer’s concern is closely connected to what we observe in our distillation ratio $\gamma$ ablation in Sec. 5.4 in the revised manuscript. When $\gamma$ is small, the residuals provides a coarse, trajectory-level bias without dominating the backward prediction. However, when $\gamma$ becomes too large, the influence of the residual becomes effectively pixel-wise in practice. In this case, the backward path is strongly pulled toward the forward path’s local structures, which can lead to locally distorted textures or over-emphasis of details.
>
> We agree with the reviewer that problematic scenarios can still occur. When the forward trajectory ends far from the ground-truth end frame, or when the scene involves new objects and large-scale rearrangements, the assumption that forward and backward latents are semantically aligned becomes weaker. In such extreme cases, the distilled residual may partially misalign with the backward latent and thus provide sub-optimal guidance. We explicitly discuss such failure cases in Appendix D of the revised manuscript, with corresponding qualitative examples.

---

### Official Review · Reviewer_CGhH · 2025-10-30

**Soundness:** 3
**Presentation:** 3
**Contribution:** 3
**Rating:** 6
**Confidence:** 5

**Summary:**

This paper addresses the issue of motion priors misalignment when adapting image-to-video diffusion models for generative in-betweening. Unlike existing adaptation methods, the proposed approach distills the forward motion prior conditioned on the first input frame into backward motion prior, eliminating the need for conditioning on the second end frame. Experiments demonstrate that this simple yet effective motion prior distillation achieves better performance compared to baseline methods.

**Strengths:**

This paper tackles a well-defined problem: the mismatch of generated motion priors that occurs when conditioning the image-to-video denoiser on each end frame during the adaptation of image-to-video models for generative in-betweening. The novel aspect of the proposed approach lies in converting the forward motion noise estimate into a backward motion noise estimate using per-frame motion residuals. This design allows the backward motion to be learned without relying on denoiser estimates conditioned on the second end frame, which could otherwise generate a complete different motion path that is difficult to align with the forward trajectory.

**Weaknesses:**

When the end point of the forward motion prior path is far from the  second input end frame,  Eq (15) would not be a good initialization. The paper should discuss this limitation and include examples to to illustrate the scenarios in which the proposed method is most effective and scenarios in which is not effective.

**Questions:**

1. GI (Wang et al., 2025b) also addresses the motion prior misalignment issue by sharing temporal self-attention maps in a time-reversed manner. Based on the video results, both GI and the proposed method appear to perform similarly for rigid motions, such as those involving cars or boats. I am curious about the specific advantages of the proposed approach over GI in practical application scenarios.

2. I also wonder how the proposed method performs when the two end frames are farther apart—for example, 50 frames. In such cases, using an image-to-video model capable of generating longer sequences (e.g., 50 frames), would the proposed method remain effective? Such a setting might further amplify the limitations discussed in the weaknesses section, as the motion priors will be more ambiguous over longer sequence.

---

> ### Author Response · Authors · 2025-11-20
>
> We sincerely appreciate the reviewer’s constructive feedback and insightful questions. The corresponding revisions in the paper are highlighted in blue-colored texts.
>
> > **W1**: When the end point of the forward motion prior path is far from the second input end frame, Eq. (15) would not be a good initialization. The paper should discuss this limitation and include examples to illustrate the scenarios in which the proposed method is most effective and scenarios in which it is not effective.
>
> **A**: We agree with your concern and suggestion. One of the key components in the proposed method is the endpoint initialization in Eq. (15) that initializes the backward noise using the latent encoding of the end frame. This implicitly assumes that the end frame of the forward path is reasonably consistent with the ground truth end frame. When the forward trajectory ends far from the ground truth end frame, Eq. (15) may become a less informative initialization. We experimentally find that our method is most effective when both keyframes contain the same object and share a semantically coherent motion. In such scenarios, the forward motion prior provides the reliable global trajectories, and distilling it into the backward path successfully reduces undesirable artifacts. Importantly, this robustness is enhanced by performing distillation at early denoising stages with additional re-noising, where the diffusion model still encodes global, low-frequency motion structure rather than fine appearance details. However, as discussed in the existing methods, our method still struggles with the inevitable problems that arise when the end frame introduces entirely new objects or undergoes large-scale scene rearrangements. In response to the reviewer’s request, we present some effective cases and some failure cases in Figs. IV to VI and Fig. III, respectively. We also added these important details in Appendix E of the revised paper.
>
> > **Q1**: Difference from GI
>
> **A**: Thank you for the valuable question. GI improves backward motion fidelity by fine-tuning SVD through rotated temporal self-attention maps. While this design encourages consistency between two paths, the two paths are still driven by different conditioning frames. Because each conditioning frame induces its own motion prior, the two paths inherently follow different motion directions and object placements, making alignment difficult even after fine-tuning. Additionally, the backward-motion network is trained only on a small collection of videos, which may not fully capture the diverse backward motion patterns. As shown in Fig. VI in the revised paper, this can lead to motion conflicts such as two surfers being generated, even though there must be one. Instead, our method reconstructs a backward estimate directly from the forward motion residuals, so that the backward path no longer introduces its own motion prior but instead follows the time-reversed motion induced by the start frame, leveraging the faithful forward motion prior of SVD. We also report a computational cost comparison in the revised paper of Appendix C, which highlights the practical advantages over GI. For a more comprehensive comparison, we further discuss the details of the differences from GI as well as another fine-tuning method (FCVG) in Appendix B of the revised paper.
>
> > **Q2**: Farther frame cases (e.g., 50-frame gaps)
>
> **A**: Following your comment, we conducted an additional experiment on large temporal gaps, where there is 50 frame-gap between the two keyframes. In this challenging scenario, both the previous time reversal sampling method (TRF) and our method exhibit the performance degradation. However, TRF severely suffers from ghosting and back-and-forth motion artifacts over our method, thereby producing less plausible intermediate frames, as shown in Fig. II of the revised paper.

---

### Author Response · Authors · 2025-11-26
**General Response to Reviewers**

Dear Reviewers,

We sincerely appreciate your valuable time and effort in reviewing our manuscript. We are glad that the reviewers appreciated the clear motivation and problem definition of our work (`CGhH`, `mj1L`, `6htp`), the novelty of our method (`CGhH`, `6htp`), the extensiveness of our experiments (`6htp`), and the improvements, especially in the zero-shot setting (`6htp`).

In response to your valuable comments, we have carefully revised the manuscript. We understand that you have numerous responsibilities, so we provide a brief summary of our main revisions below.

In the revision, we have
* explained the differences between our method and GI in Appendix B, as raised by reviewer `CGhH`.
* added discussions on effective and challenging scenarios in Appendix E, as suggested by reviewer `CGhH`.
* included additional inbetweening results for large temporal gaps in Appendix D, as raised by reviewer `CGhH`.
* provided clearer explanations of motion prior conflict in Sec. 1, addressing the concerns raised by reviewr `mj1L`.
* refined the figures (Fig. 1 and Fig. 2) to provide more detailed explanations in Sec. 1, addressing the points raised by reviewer `mj1L`.
* provided detailed explanations about early-stage applications of our method in Sec. 4.2, as requested by reviewer `6htp`.
* included additional ablation study results on the distillation ratio in Sec. 5.4, as requested by reviewer `6htp`.
* reported our method’s computational efficiency in Appendix C, as requested by reviewer `6htp`.

In the revised manuscript, changes corresponding to each reviewer are highlighted in different colors; blue for reviewer `CGhH`, green for reviewer `mj1L`, orange for reviewer `6htp`. We hope that these revisions address your concerns and help clarify the strengths of our work. If you are satisfied with the revisions, we would be very grateful if you could consider an improved score.

Thank you again for your time and insightful comments.

Best regards,

Authors

---

### Meta-Review · Area_Chair_aRUv · 2026-01-06

**Summary:**

This submission was reviewed by three expert reviewers, with the ratings of: 2 borderline accept, and 1 borderline reject. The major concerns from the reviewers are about the limitations of the technical design, differences and technical novelty when compared to closely related works, misalignment and motion conflict, unclear explanation of the stage the proposed method is applied to, study on distillation step ratios, and computational efficiency. The authors provided a rebuttal for the concerns raised by the reviewers, and there was a further discussion between the authors and one reviewer, while the other reviewers remained silent during the discussion phase.

After carefully going through the review comments, the authors' rebuttal, and the discussions, it can be seen that most major concerns and questions are addressed by the authors' further response and experiments. Within the discussion, one reviewer also acknowledged the further clarification from the authors. Although there are still some concerns outstanding, they are not major and can be addressed in a minor revision. As a result, the AC is happy to recommend acceptance of this paper, but strongly suggests that the authors incorporate all the further responses/clarifications/experiments/evidence/etc. into their final version.

**Reviewer Concerns:**

Concerns that the AC thinks were addressed by the rebuttal: the limitation of the initialization in Eq.15 and illustration examples; differences between the proposed method and GI; misalignment and motion conflict; ablation study on distillation step ratio; computational efficiency analysis.

Concerns that are still outstanding: the misleading terminology of motion conflict could be more clear throughout the whole paper; limited evidence of the early-stage application of the proposed method, though some references are provided.

**Reviewer Scores:**

According to the review comments, and the rebuttal, for each review the reviewer might have changed their score in the way below, if they had been able to participate fully in the discussion:
* Reviewer CGhH: borderline accept to accept, or unchanged
* Reviewer mj1L: borderline reject to borderline accept
* Reviewer 6htp: borderline accept unchanged, or to accept

---

### Decision · Program_Chairs · 2026-01-26

Accept (Poster)